# Feasibility and Short-Term Outcomes in Liver-First Approach: A Spanish Snapshot Study (the RENACI Project)

**DOI:** 10.3390/cancers16091676

**Published:** 2024-04-26

**Authors:** Mario Serradilla-Martín, Celia Villodre, Laia Falgueras-Verdaguer, Natalia Zambudio-Carroll, José T. Castell-Gómez, Juan L. Blas-Laina, Vicente Borrego-Estella, Carlos Domingo-del-Pozo, Gabriel García-Plaza, Francisco J. González-Rodríguez, Eva M. Montalvá-Orón, Ángel Moya-Herraiz, Sandra Paterna-López, Miguel A. Suárez-Muñoz, Maialen Alkorta-Zuloaga, Gerardo Blanco-Fernández, Enrique Dabán-Collado, Miguel A. Gómez-Bravo, José I. Miota-de-Llamas, Fernando Rotellar, Belinda Sánchez-Pérez, Santiago Sánchez-Cabús, David Pacheco-Sánchez, Juan C. Rodríguez-Sanjuan, María A. Varona-Bosque, Lucía Carrión-Álvarez, Sofía de la Serna-Esteban, Cristina Dopazo, Elena Martín-Pérez, David Martínez-Cecilia, María J. Castro-Santiago, Dimitri Dorcaratto, Marta L. Gutiérrez-Díaz, José M. Asencio-Pascual, Fernando Burdío-Pinilla, Roberto Carracedo-Iglesias, Alfredo Escartín-Arias, Benedetto Ielpo, Gonzalo Rodríguez-Laiz, Andrés Valdivieso-López, Emilio De-Vicente-López, Vicente Alonso-Orduña, José M. Ramia

**Affiliations:** 1Department of Surgery, Hospital Universitario Virgen de las Nieves, 18014 Granada, Spain; natalia.zambudio.sspa@juantadeandalucia.es; 2Instituto de Investigación Biosanitaria ibs.GRANADA, 18012 Granada, Spain; 3Department of Surgery, School of Medicine, University of Granada, 18016 Granada, Spain; 4Department of Surgery, Hospital General Universitario Dr. Balmis, 03010 Alicante, Spain; cvillodre@umh.es (C.V.); ramia_jos@gva.es (J.M.R.); 5ISABIAL, Instituto de Investigación Sanitaria y Biomédica de Alicante, 03010 Alicante, Spain; 6Department of Surgery, Universidad Miguel Hernández, 03202 Alicante, Spain; 7Department of Surgery, Hospital Universitario Dr. Josep Trueta, 17007 Girona, Spain; lfalgueras.girona.ics@gencat.cat; 8Department of Surgery, Hospital Universitario La Paz, 28046 Madrid, Spain; jtcastell@quironsalud.es; 9Department of Surgery, Hospital Royo Villanova, 50015 Zaragoza, Spain; jlblas@salud.aragon.es; 10Department of Surgery, Hospital Universitario Lozano Blesa, 50009 Zaragoza, Spain; vmborrego@salud.aragon.es; 11Department of Surgery, Hospital Universitario Dr. Peset, 46017 Valencia, Spain; domingo_cardel@gva.es; 12Department of Surgery, Hospital Universitario Insular, 35016 Las Palmas de Gran Canaria, Spain; ggarpla@gobiernodecanarias.org; 13Department of Surgery, Hospital Clínico Universitario de Santiago, 15706 Santiago de Compostela, Spain; francisco.javier.gonzalez.rodriguez2@sergas.es; 14Department of Surgery, Hospital Universitario y Politécnico La Fe, IIS La Fe, Ciberehd ISCIII, 46026 Valencia, Spain; montalva_eva@gva.es; 15Department of Surgery, Hospital Universitario de Castellón, 12004 Castelló de la Plana, Spain; moya_ang@gva.es; 16Department of Surgery, Hospital Universitario Miguel Servet, 50009 Zaragoza, Spain; spaterna@salud.aragon.es; 17Department of Surgery, Hospital Universitario Virgen de la Victoria, 29010 Málaga, Spain; mangel.suarez.sspa@juantadeandalucia.es; 18Department of Surgery, Hospital Universitario Donostia, 20014 San Sebastián, Spain; maialen.alkortazuloaga@osakidetza.eus; 19Department of Surgery, Hospital Universitario de Badajoz, 06006 Badajoz, Spain; gerardoblanco@unex.es; 20Department of Surgery, Hospital Universitario San Cecilio, 18016 Granada, Spain; enriquej.daban.sspa@juntadeandalucia.es; 21Department of Surgery, Hospital Universitario Virgen del Rocío, 41013 Sevilla, Spain; mangel.gomez.sspa@juntadeandalucia.es; 22Department of Surgery, Hospital Universitario de Albacete, 02006 Albacete, Spain; jimiotad@sescam.jccm.es; 23Department of Surgery, Clínica Universidad de Navarra, 31008 Pamplona, Spain; frotellar@unav.es; 24Department of Surgery, Hospital Regional Universitario de Málaga, 29010 Málaga, Spain; belinda.sanchez.sspa@juntadeandalucia.es; 25Department of Surgery, Hospital Universitario de la Santa Creu i Sant Pau, 08041 Barcelona, Spain; ssanchezca@santpau.cat; 26Department of Surgery, Hospital Universitario Río Hortega, 47012 Valladolid, Spain; dpachecosa@saludcastillayleon.es; 27Department of Surgery, Hospital Universitario Marqués de Valdecilla, 39008 Santander, Spain; juancarlos.rodriguezs@scsalud.es; 28Department of Surgery, Hospital Universitario Nuestra Señora de la Candelaria, 38010 Santa Cruz de Tenerife, Spain; mvarbosa@gobiernodecanarias.org; 29Department of Surgery, Hospital Universitario de Fuenlabrada, 28942 Madrid, Spain; lucia.carrion@salud.madrid.org; 30Department of Surgery, Hospital Clínico Universitario, 28040 Madrid, Spain; sofiacristinadela.serna@salud.madrid.org; 31Department of Surgery, Hospital Universitario Vall d’Hebron, 08035 Barcelona, Spain; cristina.dopazo@vallhebron.cat; 32Department of Surgery, Hospital Universitario La Princesa, 28006 Madrid, Spain; elena.perez@uam.es (E.M.-P.); dmcecilia@salud.madrid.org (D.M.-C.); 33Department of Surgery, Hospital Universitario Virgen de la Salud, 45004 Toledo, Spain; 34Department of Surgery, Hospital Universitario Puerta del Mar, 11009 Cádiz, Spain; mariaj.castro.santiago.sspa@juntadeandalucia.es; 35Department of Surgery, Hospital Clínico Universitario, 46010 Valencia, Spain; dorcaratto_dim@gva.es; 36Department of Surgery, Hospital Quirón, 50006 Zaragoza, Spain; martalgutierrezdi@salud.aragon.es; 37Department of Surgery, Hospital Universitario Gregorio Marañón, 28007 Madrid, Spain; josemanuel.asencio@salud.madrid.org; 38Department of Surgery, Hospital Universitario del Mar, 08003 Barcelona, Spain; fburdio@psmar.cat (F.B.-P.); bielpo@psmar.cat (B.I.); 39Department of Surgery, Hospital Universitario Álvaro Cunqueiro, 36312 Vigo, Spain; roberto.carracedo.iglesias@sergas.es; 40Department of Surgery, Hospital Universitario Arnau de Vilanova, 25198 Lleida, Spain; aescartin.lleida.ics@gencat.cat; 41Department of Surgery, Hospital Universitario de León, 24008 León, Spain; 42Department of Surgery, Hospital Universitario de Cruces, 48903 Barakaldo, Spain; acanvalecha@telefonica.net; 43Department of Surgery, Hospital Universitario HM Sanchinarro, 28050 Madrid, Spain; correo@emiliovicente.es; 44Department of Medical Oncology, Hospital Universitario Miguel Servet, 50009 Zaragoza, Spain; valonsoo@salud.aragon.es

**Keywords:** colorectal cancer, liver metastases, liver-first approach, disease-free survival

## Abstract

**Simple Summary:**

Current evidence does not provide enough information for selecting a tailored approach pathway in patients with colorectal cancer and synchronous liver metastases. There are no randomized clinical trials or prospective series comparing the classical approach with the liver-first approach. In addition, information on the proportion of patients who actually complete the therapeutic regimen is limited. The RENACI Project was a prospective National Registry performed on patients with colorectal cancer and synchronous liver metastases undergoing the liver-first approach. This study aimed to present the data of feasibility and short-term outcomes of the Spanish National Registry of Liver First Approach (the RENACI Project).

**Abstract:**

(1) Background: The liver-first approach may be indicated for colorectal cancer patients with synchronous liver metastases to whom preoperative chemotherapy opens a potential window in which liver resection may be undertaken. This study aims to present the data of feasibility and short-term outcomes in the liver-first approach. (2) Methods: A prospective observational study was performed in Spanish hospitals that had a medium/high-volume of HPB surgeries from 1 June 2019 to 31 August 2020. (3) Results: In total, 40 hospitals participated, including a total of 2288 hepatectomies, 1350 for colorectal liver metastases, 150 of them (11.1%) using the liver-first approach, 63 (42.0%) in hospitals performing <50 hepatectomies/year. The proportion of patients as ASA III was significantly higher in centers performing ≥50 hepatectomies/year (difference: 18.9%; *p* = 0.0213). In 81.1% of the cases, the primary tumor was in the rectum or sigmoid colon. In total, 40% of the patients underwent major hepatectomies. The surgical approach was open surgery in 87 (58.0%) patients. Resection margins were R0 in 78.5% of the patients. In total, 40 (26.7%) patients had complications after the liver resection and 36 (27.3%) had complications after the primary resection. One-hundred and thirty-two (89.3%) patients completed the therapeutic regime. (4) Conclusions: There were no differences in the surgical outcomes between the centers performing <50 and ≥50 hepatectomies/year. Further analysis evaluating factors associated with clinical outcomes and determining the best candidates for this approach will be subsequently conducted.

## 1. Introduction

Colorectal cancer (CRC) is considered the second most common malignancy worldwide, with approximately 15–20% of cases presenting synchronous liver metastases (SCRLM) at time of diagnosis [1,2,3]. Surgical resection, often in combination with chemotherapy, may offer long-term survival in a significant proportion of patients [4].

Resection of both the primary tumor and liver metastases may offer a real chance for cure, but it is possible only for a minority of patients. Although different strategies have been used in the past, the current trend, proposed by Mentha et al. [5], for patients with asymptomatic colorectal tumors with initially unresectable or borderline resectable liver metastases, lies in performing high-impact chemotherapy first, resection of liver metastases second, followed by chemo/radiotherapy of the primary tumor in case of rectal tumors, and finally removal of the primary tumor. This strategy is also called the reverse strategy or liver-first approach (LFA).

It has been suggested that LFA may be particularly indicated for colorectal cancer patients with SCRLM to whom preoperative chemotherapy treatment opens a potential “window” in which liver resection may be undertaken [6,7]. However, the surgical strategy should be decided according to the hepatic tumor burden [8].

Another strategy entails primary CRC and liver metastases resection in a single operation (simultaneous strategy) [9], although simultaneous resection did not show better survival, while was associated with more complications [4].

Baltatzis et al. [9], in a systematic review and metanalysis, compared these techniques, namely sequential primary-first, LFA, or synchronous resection. Besides the potential bias and differences in study protocols, this study did not find differences in major complications, post-operative death and 5-year survival among the three techniques [9]. Additionally, there were no differences in disease recurrence among these techniques [9]. Similarly, Salvador-Rosés et al. [10] did not find significant differences in the complete resection rate between the primary-first and the LFA strategies, although both strategies were feasible and safe.

Moreover, the results of a meta-analysis that compared the perioperative outcomes of LFA and classical strategy for the management of SCLRM did not find significant differences in clinical outcomes between these techniques. Nevertheless, it suggested that LFA may be a better option for patients with a higher burden of liver disease, while the classical strategy may be a valuable option for patients who do not require a downstaging therapy [11].

Current evidence does not provide enough information for selecting a tailored approach pathway in patients with CRC and SCRLM [6,7,8,9,10,11,12,13,14,15,16,17,18,19]. To the authors’ knowledge, there are no randomized clinical trials or prospective series comparing the classical approach with LFA. In addition, information on the proportion of patients who actually complete the therapeutic regimen is limited.

Although randomized controlled trials represent the highest hierarchical level of evidence, they are not immune to flaws [20]. They require strict inclusion and exclusion criteria, thus limiting the generalizability of the findings to broader populations [21]. In recent years, prospective clinical registries have been increasingly recognized as a valuable tool for improving the value of healthcare via the use of outcome data [21].

On the other hand, there is an inverse relationship between hospital and surgeon volume and mortality in many types of complex surgery.

The RENACI Project was a prospective National Registry performed of patients with CRC and SCRLM undergoing LFA. This study aimed to present the data of feasibility and short-term outcomes of the Spanish National Registry of Liver First Approach (the RENACI Project).

## 2. Materials and Methods

### 2.1. Design

We performed a prospective and observational study conducted on consecutive patients with CRC and SCRLM (defined as presence of liver metastases at the time of colorectal cancer diagnosis) recruited from the Hepato-Pancreato-Biliary (HPB) Units of Spanish hospitals from 1 June 2019 to 30 August 2020. The study coordinators contacted by email the coordinator of the HPB Surgery of all the Spanish hospitals that perform liver surgery. A total of 40 second (area hospitals with approximately 500 beds, and on average of 270 specialists and 50 residents) and third-level (university reference hospitals with approximately 800–1000 beds, on average of 680 specialists and 300 residents, and great teaching intensity) hospitals decided to participate in the study.

The study protocol was approved by the Ethics Committee of Aragon on 27 May 2019 (C.P.–C.I. PI19/256); Clinical Trials registry: NCT04683783. All patients were fully informed about the details of the study, and patients provided written informed consent at the beginning of the study. The ethical principles outlined in the Declaration of Helsinki and Good Clinical Practice were followed.

### 2.2. The RENACI Project

The RENACI Project includes data from 40 second-level and third-level Spanish hospitals. The objective is to recruit patients prospectively during a period of one year (extended three more months due to the COVID-19 pandemic), to analyze the feasibility of LFA, postoperative short-term and long-term outcomes, and long-term overall survival and disease-free survival.

### 2.3. Patients

Consecutive patients with a clinical diagnosed of CRC and SCRLM, who underwent a LFA during the study period, in any of the Spanish centers participating in the study, and that met the inclusion criteria were included.

### 2.4. Inclusion/Exclusion Criteria

Male and female subjects aged ≥18 years, based on an American Society of Anaesthesiologists (ASA) physical status classification system [22] score I-III, who were selected for scheduled surgery for CRC with SCRLM using the LFA, were included in the study.

Patients were excluded if they were <18 years, had an ASA score ≥4, had undergone urgent surgery, showed unwillingness to comply with the investigators and protocol indications, or were incapable of providing written consent or did not sign the consent form. Patients with extrahepatic disease were also excluded.

Each participating center meticulously adhered to these inclusion criteria throughout the study.

### 2.5. Treatment Strategy

LFA was initially described for asymptomatic colorectal tumors with unresectable or potentially resectable synchronous liver metastases. In those patients with partial response or stabilization of liver disease, liver surgery was performed to prioritize the removal of the most prognostically relevant disease (liver metastases). In cases of locally advanced rectal tumors, radiotherapy or chemotherapy/radiotherapy was carried out, and finally surgery of the primary tumor was performed.

### 2.6. Outcomes

The primary end-point was the percentage of patients who complete the treatment paradigm: neoadjuvant chemotherapy + liver surgery ± chemotherapy/radiotherapy of the primary tumor + surgery of the primary tumor.

The secondary end-points were 90-day postoperative morbidity, including liver and colorectal surgery (all type of postoperative complications), and to investigate the volume effect on outcomes this complex surgery.

### 2.7. Study Variables

The following variables were studied: age, sex, Body Mass Index, ASA grade, and past medical history; clinical symptoms; carcinoembryonic antigen (CEA) and carbohydrate antigen (CA) 19.9 preoperative levels; location of the primary tumor, number, size, and location of liver metastases; need for stent placement or colostomy, neoadjuvant chemotherapy, and time from diagnosis to start of chemotherapy; portal embolization, two-stage hepatectomy, type of surgery, major (greater than or equal to three segments) and minor (less than three segments) hepatectomy, operating time, approach, intraoperative blood loss, clamping time, R status, degree of tumor regression, postoperative morbidity and mortality (according to the Clavien–Dindo classification) [23], bile leak, post-hepatectomy insufficiency and hemorrhage defined by International Study Group of Liver Surgery classification [24,25,26], length of hospital stay (LOS), readmissions, adjuvant chemotherapy, and radiotherapy; number of patients with resection of the primary tumor, type of surgery, approach, operating time, intraoperative blood loss, postoperative morbidity and mortality after primary resection, LOS, readmissions, histological type, TNM classification, degree of tumor regression, and adjuvant chemotherapy; and postoperative follow-up (months), death, and recurrence.

### 2.8. Statistical Analysis

The statistical analysis was performed using R software (version 4.0.3) (https://www.R-project.org/, accessed on 11 October 2023). Descriptive statistics number (percentage), mean (95% confidence interval, CI), or median (interquartile range, IQR) were used as appropriate.

Depending on the number of cases provided by each collaborating hospital, a retrospective analysis was carried out to detect the power of the differences observed in the data. The variables of interest were described in univariate and bivariate tables according to the study groups. For comparisons between groups, parametric (t-student, ANOVA) and non-parametric (Mann–Whitney, Kruskal–Wallis) tests were used on continuous variables depending on their distribution and Fisher or chi-square tests for categorical variables.

The study sample was divided according to the number of hepatectomies/year. Subjects operated on in centers that performed <50 hepatectomies per year were compared with cases operated on in centers that performed ≥50 hepatectomies per year.

To investigate the relationship between different variables, correlation analysis and/or bivariate or multivariate linear and logistic regression were used. In addition, the longitudinal variation of certain variables of interest were studied, for which Kaplan–Meier estimators and bi- or multi-variate analysis using Cox models were carried out.

## 3. Results

### 3.1. General Information

A total of 40 hospitals of the 72 centers contacted agreed to participate in the project. Of the 40 participating centers, liver transplantation was performed in 16 (40%) hospitals.

Throughout the study inclusion period, a total of 2288 hepatectomies were performed in the study centers, 1350 for CRLM, with a mean of 57.2 hepatectomies per center (23 to 112). Among them, 150 (11.1%) patients had undergone a LFA and were included in the study. In total, 63 (42.0%) LFAs were performed in centers that performed <50 hepatectomies per year, and 87 (58.0%) LFAs were performed in centers that performed ≥50 hepatectomies per year.

### 3.2. Preoperative Clinical and Demographic Characteristics

The mean (95% CI) age was 61.9 (52.4 to 69.1) years, and 96 (64.0%) patients were men. The mean body mass index (BMI) was 25.9 (23.8 to 29.5) kg/m^2^, with 32 (21.5%) patients considered to be obese (BMI ≥ 30 kg/m^2^). Regarding ASA, 9 (6.0%), 77 (51.3%) and 64 (42.7%) were classified as ASA I, II, and III, respectively. The proportion of operated patients classified as ASA III was significantly higher in centers that perform ≥50 hepatectomies per year (difference: 18.9%; 95% CI: 2.9% to 33.3%; *p* = 0.0213).

Table 1 shows the main demographic and clinical characteristics of the study population.

There were no differences between both groups in terms of preoperative location of liver metastases or bilobar involvement (30 patients [47.6%] in centers < 50 hepatectomies/year vs. 55 patients [63.2%] in centers ≥ 50 hepatectomies/year, *p* = 0.083).

The most frequent symptoms were rectal bleeding (41.3%; 62/150) and abdominal pain (20.7%; 31/150), while 25 (16.7%) patients were asymptomatic (Table 1). Preoperative mean carcinoembryonic antigen (CEA) was 10.3 (3.84 to 51.5) ng/mL (Table 1). KRAS gene mutation was observed in 43 (30.3%) patients (Table 1). All of the patients received neoadyuvant chemotherapy, the majority based on the FOLFOX and FOLFIRI regimens in combination with monoclonal antibodies (55.3%), with a mean of 6 cycles.

### 3.3. Liver Resection Procedure

Nineteen (12.8%) patients had undergone a previous portal vein embolization and underwent a 2-stage hepatectomy. Regarding the type of surgery, 53 (35.6%) patients underwent segmentectomy ± radiofrequency ablation (RF), 41 (27.5%) patients underwent right hepatectomy ± wedge resection ± RF, and 36 (24.2%) patients underwent wedge resection ± RF. The surgical approach was open surgery in 87 (58.0%) patients and laparoscopic surgery in 54 (36.0%) patients (Table 2). Resection margins were R0 in 117 (78.5%) patients and R1 in 22 (14.8%) patients. The mean surgical time of liver resection in the overall population was 240 (186 to 305) minutes (Table 2).

### 3.4. Characteristics of the Primary Tumour Surgical Procedure

In total, 70 (46.6%) received chemotherapy between liver resection and primary surgery and 34 (22.8%) patients received radiotherapy. Primary surgery was performed using the laparoscopic approach in 87 (66.4%) patients, and it was not possible to perform colorectal cancer surgery in 16 (10.7%) patients (Table 3) for the following reasons: 9 due to complications after liver surgery, 6 due to progression of liver disease, and 1 due to postoperative death. The median time between both interventions was 2.14 months, without differences between both groups (2.09 vs. 2.14, *p* = 0.356).

One-hundred and thirty-four (89.3%) patients completed the therapeutic regime (neoadjuvant chemotherapy + liver resection ± chemotherapy/radiotherapy of the primary tumor + surgery of the primary tumor). In other words, the overall feasibility was 89.3%.

### 3.5. Safety

Regarding the safety profile, 40 (26.7%) patients had complications after the liver resection and 36 (27.3%) patients had complications after the primary tumor procedure. No significant differences were found in both the liver resection and primary tumor procedure between the centers that performed <50 hepatectomies per year and those that performed ≥50 hepatectomies per year (Table 4 and Table 5).

Regarding liver resection, five (3.4%) patients required a reintervention, one (0.7%) patient required a percutaneous drainage, and four (27%) patients required a surgical reintervention. There were no significant differences in the complication rates between patients with 2-stage hepatectomy and the rest of the patients (28.1 vs. 24.3, *p* = 0.682), whereas in the primary tumor surgical procedure, nine (6.9%) patients required a surgical reintervention.

In the overall study sample, the mean hospital stay was 6 (4.0 to 9.0) days and 7 (5.0 to 10.0) days for the liver resection and the primary tumor surgery, respectively (Table 4 and Table 5).

## 4. Discussion

The results of the current study showed that 134 (89.3%) patients completed the therapeutic regime. Additionally, 39 (26.0%) patients and 35 patients (26.5%) presented complications after liver resection and primary tumor surgery, respectively, with no significant differences between the centers that performed <50 hepatectomies per year and those that performed ≥50 hepatectomies per year.

Our study also showed that liver-first strategy rates in Spain (11.1%) are in line with the current figures reported worldwide (approximately 13%) [8].

An interesting point, in our opinion, is that 60% of patients had undergone minor liver surgery (either segmentectomies or wedge resections), whereas 40% underwent major hepatectomies. Although, at first, this may seem like a contradiction (it would be expected to resect larger ones, since these are livers with a greater tumor load), it is in line with the worldwide LiverMet Survey registry data, where the proportion of major hepatectomies was 40% [8].

To our knowledge, this is the largest prospective series analyzed so far that evaluates the data of feasibility in patients with CRC and SCRLM who underwent LFA.

The LFA was originally described for colorectal tumors with unresectable or resectable-borderline metastases, but its indications have gradually expanded. Several factors, including the improvements in chemotherapy, the appearance of newer biological agents, such as bevacizumab and cetuximab [27,28], as well as advances in the availability of liver surgery, anesthesia, and critical care, have made liver-first strategy a feasible option for patients with SCRLM [6,7,8,9,10,11,12,13,14,15,16,17,18,19].

The rationale behind the LFA is mainly based on two pillars: performing early liver resection allows control of SCRLM, which may increase the chance of curative surgery; and the subsequent primary tumor surgery may prevent loss of primary tumor induced inhibition of the metastases [29]. According to the results of meta-analysis recently published, as compared to simultaneous approach, LFA was associated with lower risk of postoperative mortality, but with a longer length of stay [30].

Most of evidence that evaluated this strategy only included patients with liver resection, but did not provide data on the primary surgical procedure [6,7,8,9,10,11,12,13,14,15,16,17,18,19]. Therefore, there is little data in the scientific literature on how many patients scheduled for this strategy complete both surgeries and/or undergo the full chemo/radiation therapy.

Currently available evidence has not clarified the role of LFA in SCRLM and its oncologic superiority over the other strategies is still to be proven [6,7,8,9,10,11,12,13,14,15,16,17,18,19,29,30,31]. Moreover, current evidence points in the same general direction indicating neither inferiority nor superiority of the LFA versus the primary-first approach [6,7,8,9,10,11,12,13,14,15,16,17,18,19,29,30,31].

Our study is not focused on comparing the different techniques, but rather in evaluating the feasibility and safety of the LFA, evaluating the proportion of patients who really are able to follow this treatment paradigm.

In our study, 134 (89.3%) patients completed the liver-first therapeutic regime. These figures seem to slightly greater than the 76.1% (70/92) of patients reported by de Jong et al. [15], but similar to the 88.9% (16/18) of patients found by Wang et al. [32], although they evaluated a significantly lower number of cases. Additionally, the feasibility rate of the current study seems to be greater than that reported by two systematic reviews [33,34] and different small series [32,35,36,37,38,39,40] (see Table 6).

As compared to Giuliante et al. [8], the overall morbidity was similar (30.4% versus approximately 27%, respectively), although our study was prospective, which is usually associated with a higher rate of complications. Among our patients, overall postoperative morbidity was 26.7% following liver resection and 27.6% after primary tumor surgical procedure. The rates of major complications (Clavien ≥ IIIa) were 12.0% (18/150 patients) and 12.1% (16/132 patients) in the liver and primary-first approach, respectively. In total, 1 (0.7%) patient died in LFA group versus 0 (0.0%) in the primary-first one. These data were similar to that reported by other authors [15,32,34,43].

However, our study did not find significant differences between the centers performing <50 and ≥50 hepatectomies/year. Simultaneous resection tends to have a high completion rate, but has been associated with heightened risks of complications [4]. Therefore, safety-centered approaches would be recommended for facilities performing fewer than 50 liver resections annually.

Interestingly, our series shows that approximately 50% of patients did not have a rectal tumor, which clearly suggests that liver-first strategy is expanding its indications.

However, LFA has preferentially been applied to patients with rectal tumors and high liver tumor burden [6,7,8,9]. In patients with CRC and liver metastases, both resections can be performed in a single procedure [9]. Interestingly, this strategy did not have better survival outcomes, while it was associated with more complications [4]. Current evidence suggests that in patients with CRC, LFA is not inferior to other approaches in patients with unilobar SCRLM [8,9]. Nevertheless, LFA was associated with a clear survival advantage over both the primary-first and simultaneous approaches in patients with multiple bilobar metastases [8,9].

Finally, it should be mentioned that despite LFA strategy prioritizes the removal of metastases, it still includes a chemotherapy-free period of at least 3 months after liver surgery [6,7]. It has been recently proposed a new LFA strategy that proposed resection of the liver metastases during the interval between long-course chemoradiation and rectal cancer surgery [44]. The authors reported that 87.5% of patients successfully underwent the liver-first strategy and underwent both liver and rectal treatment [44]. These results are similar to those found in our study, with the particularity that our study included 150 cases and the study by Bonnet et al. [44] only included 24 patients.

Nevertheless, this strategy offers interesting possibilities that must be analyzed in future studies with a larger number of cases.

The current work has several limitations that should be taken into consideration when interpreting its results. As this is a multicenter study, there may be some differences between the surgical techniques between the different centers and may influence surgical outcomes. Likewise, in a multicenter study of these characteristics, without a specific definition of what is unresectable or borderline resectable, there may be disparate criteria in this sense, depending on the experience of the surgical team, which represents another limitation. However, we clearly defined the standard procedure and the limits on acceptable technical variation. The lack of comparison of our cohort of patients with those who underwent bowel-first and simultaneous resection is a limitation to support the feasibility of the LFA. This study was focus on describing the characteristics of the study sample and provided only preliminary results. Nevertheless, further analysis evaluating the association between potential relevant clinicopathological factors and prognosis, determining the best candidates for LFA, will be performed. Additionally, these new analyses might open the door to the development of new and different therapeutic algorithms and to define expert levels in liver surgery.

Its main strengths are its prospective design and the fact that it reflects the management, in a real-world scenario, of the CRC with SCRLM surgical approach in Spain.

## 5. Conclusions

The Spanish National Registry of Liver First Approach (RENACI) project was one the largest multicentre clinical studies to prospectively evaluate the feasibility of LFA in patients with colorectal cancer and SCRLM at the time of diagnosis.

According to our results, 89.3% of the patients completed the entire therapeutic paradigm. Additionally, our series found an overall morbidity rate of 26.0% and 26.5% following liver resection and after primary tumor surgical procedure, respectively. The fact that there were no differences in either the type of results or the surgical outcomes between the centers that do <50 hepatectomies per year and those that perform ≥50 hepatectomies per year highlights the high degree of expertise of all the surgical teams that make up the RENACI database. Further analysis evaluating factors associated with clinical outcomes and determining the best candidates for this approach will be subsequently conducted.

## Figures and Tables

**Table 1 cancers-16-01676-t001:** Preoperative demographic and clinical characteristics of study sample.

	Overall Study Sample *N* = 150	Centers < 50 Hep/Year *N* = 63	Centers ≥ 50 Hep/Year *N* = 87	*p*	*N*
Age, years					
Mean [95% CI]	61.9 (52.4 to 69.1)	61.7 [53.4 to 68.1]	62.3 [52.1 to 69.4]	0.615	150
Sex, *n* (%)					
Men	96 (64.0)	38 (60.3)	58 (66.7)	0.530	150
Women	54 (36.0)	25 (39.7)	29 (33.3)		
BMI, Kg/m^2^					
Mean [95%CI]	25.9 (23.8 to 29.5)	25.4 [23.1 to 29.3]	26.3 [24.1 to 29.7]	0.500	149
Obesity	32 (21.5%)	14 (22.2%)	18 (20.9%)	1.000	149
ASA *n* (%)					
I	9 (6.0)	8 (12.7)	1 (1.15)	0.003	150
II	77 (51.3)	35 (55.6)	42 (48.3)		
III	64 (42.7)	20 (31.7)	44 (50.6)		
Location of LM, *n* (%)					
Segment I	6 (4.0)	1 (1.6)	5 (5.8)	0.402	150
Segment II	48 (32.0)	18 (28.6)	30 (34.5)	0.556	
Segment III	49 (32.7)	15 (23.8)	34 (39.1)	0.073	
Segment IVa	43 (28.7)	18 (28.6)	25 (28.7)	1.000	
Segment IVb	45 (30.0)	16 (25.4)	29 (33.3)	0.386	
Segment V	74 (49.3)	35 (55.6)	39 (44.8)	0.258	
Segment VI	80 (53.3)	34 (54.0)	46 (52.9)	1.000	
Segment VII	80 (53.3)	30 (47.6)	50 (57.5)	0.304	
Segment VIII	83 (55.3)	32 (50.8)	51 (58.6)	0.432	
Left lobe	104 (69.3)	37 (58.7)	67 (77.0)	0.027	
Right lobe	130 (86.7)	55 (87.3)	75 (86.2)	1.000	
Bilobar involvement	85 (56.7)	30 (47.6)	55 (63.2)	0.083	
NPLM					
Mean [95%CI]	3.00 [2.00 to 6.00]	3.00 [2.00 to 6.00]	4.00 [1.50 to 6.00]	0.409	148
LMS, mm					
Mean [95% CI]	30.0 [19.5;55.5]	26.5 [16.1 to 48.8]	31.5 [20.0 to 60.0]	0.226	148
Symptoms *^,1^ *n* (%)					
Anaemia	19 (12.7)	10 (15.9)	9 (10.3)	0.450	150
Asymptomatic	25 (16.7)	8 (12.7)	17 (19.5)	0.375	
Abdominal pain	31 (20.7)	10 (15.9)	21 (24.1)	0.303	
Constipation	23 (15.3)	5 (7.9)	18 (20.7)	0.056	
Rectal bleeding	62 (41.3)	30 (47.6)	32 (36.8)	0.245	
Obstruction	4 (2.7)	3 (4.8)	1 (1.6)	0.310	
Constitutional syndrome	27 (18.0)	13 (20.6)	14 (16.1)	0.617	
KRAS gene mutation	43 (31.0%)	23	20	0.459	145
Preoperative CEA (ng/mL)					
Mean [95% CI]	10.3 [3.84 to 51.5]	9.2 [3.84 to 49.27]	10.1 [2.45 to 51.5]	0.863	148
Neoadjuvant CT, *n* (%)					
FOLFIRI + Cetuximab	1 (0.7)	0 (0)	1 (1.1)	0.874	143
FOLFIRI + Panitumumab	2 (1.4)	1 (1.5)	1 (1.1)	0.468	
FOLFOX + Bevacizumab	29 (20.3)	13 (19.4)	16 (18.4)	0.926	
FOLFOX + Cetuximab	18 (12.6)	8 (11.9)	10 (11.5)	0.296	
FOLFOX + Panitumumab	19 (20.3)	13 (19.4)	16 (18.4)	0.733	
FOLFOXIRI + Bevacizumab	4 (2.8)	1 (1.5)	3 (3.4)	0.629	
XELOX + Bevacizumab	13 (9.1)	5 (7.5)	8 (9.2)	0.595	
Other	47 (32)	22 (32.8)	25 (28.7)	0.728	
Number of CT cycles					
Mean [95% CI]	6.00 [4.00 to 8.00]	6.00 [4.00 to 7.00]	6.00 [4.00 to 8.00]	0.739	143

* Princeps symptom that caused the initial visit. ^1^ Total percentage may be greater than 100%. Hep: Hepatectomy; CI: Confidence interval; BMI: Body mass index; ASA: American Society of Anesthesiologists physical status classification; LM: Liver metastases; NPLM: Number of preoperative liver metastases; LMS: Largest metastases size; CEA: carcinoembryonic antigen; CA 19.9: Carbohydrate antigen 19.9; CT: Chemotherapy.

**Table 2 cancers-16-01676-t002:** Overview of the main characteristics of the liver resection procedure.

	Overall Study Sample *N* = 150	Centers < 50 Hep/Year *N* = 63	Centers ≥ 50 Hep/Year *N* = 87	*p*	*N*
Previous portal vein embolization, *n* (%)	19 (12.7)	9 (14.3)	10 (11.5)	0.796	150
Two-stage hepatectomy, *n* (%)	19 (12.8)	7 (11.3)	12 (14.0)	0.819	148
Type of surgery, *n* (%)					
Right hepatectomy +/− wedge +/− RF	41 (27.5)	17 (27.0)	24 (27.9)	0.597	149
Left hepatectomy +/− wedge +/− RF	14 (9.4)	6 (9.5)	8 (9.3)		
Segmentectomy +/− RF	53 (35.6)	22 (34.9)	31 (36.0)		
Right trisectionectomy +/− wedge +/− RF	3 (2.0)	0 (0.0)	3 (3.5)		
Left trisectionectomy +/− wedge +/− RF	2 (1.3)	0 (0.0)	2 (2.3)		
Wedge +/− RF	37 (24.7)	18 (28.6)	19 (21.8)		
Number of wedge resection					
Mean [95% CI]	2.00 [1.00;3.75]	2.0 [1.0 to 3.0]	2.0 [1.0 to 3.8]	0.602	36
Number of segmentectomies					
Mean [95% CI]	2.00 [1.00;2.00]	2.0 [1.0 to 2.0]	2.0 [1.0 to 2.5]	0.451	53
Surgical time, minutes					
Mean [95% CI]	240 [181;308]	240 [186 to 305]	240 [180 to 300]	0.523	139
Surgical approach, *n* (%)					
Open surgery	87 (58.0)	33 (52.4)	54 (62.1)	0.232	150
Conversion	9 (6.0)	6 (9.52)	3 (3.5)		
Laparoscopic	54 (36.0)	24 (38.1)	30 (34.5)		
Bleeding (mL)					
Mean [95% CI]	200 [100;400]	300 [100 to 400]	200 [100;400]	0.300	12
Blood units transfused, *n* (%)					
0	123 (82.0)	50 (79.4)	73 (83.9)	0.490	150
1	14 (9.3)	6 (9.5)	8 (9.2)		
2	9 (6.0)	4 (6.4)	5 (5.8)		
3	2 (1.3)	2 (3.17)	0 (0.0)		
4	1 (0.7)	0 (0.0)	1 (1.6)		
>4	1 (0.7)	1 (1.6)	0 (0.0)		
Clamping time (minutes)					
Mean [95% CI]	30.0 [12.0;51.0]	30.0 [10.5 to 50.0]	30.5 [14.8 to 60.0]	0.353	146
Type of resection, *n* (%)					
R0	117 (78.5)	49 (77.8)	68 (79.1)	0.297	149
R1	22 (14.8)	12 (19.0)	10 (11.6)		
Vascular R1	9 (6.0)	2 (3.2)	7 (8.14)		
R2	1 (0.7)	0 (0.0)	1 (1.16)		
Degree of tumor regression, *n* (%)					
Grade 1	18 (17.1)	11 (22.0)	7 (12.7)	0.580	105
Grade 2	28 (26.7)	12 (24.0)	16 (29.1)		
Grade 3	24 (22.9)	13 (26.0)	11 (20.0)		
Grade 4	16 (15.2)	6 (12.0)	10 (18.2)		
Grade 0	19 (18.1)	8 (16.0)	11 (20.0)		

Hep: Hepatectomy; RF: Radiofrequency; CI: Confidence interval.

**Table 3 cancers-16-01676-t003:** Overview of the main characteristics of the primary tumor surgical procedure.

	Overall Study Sample *N* = 134	*N*
Type of resection, *n* (%)		
Abdominoperineal amputation	13 (9.9)	134
Subtotal colectomy	1 (0.8)	
Hartmann’s procedure	7 (5.3)	
Extended right hemicolectomy	2 (1.5)	
Right hemicolectomy	10 (7.5)	
Left hemicolectomy	15 (11.4)	
Exploratory laparotomy	2 (1.5)	
Anterior resection	17 (12.9)	
Lower anterior resection	33 (24.6)	
Sigmoidectomy	34 (25.4)	
Surgical time (minutes)		
Mean [95% CI]	200 [155;240]	117
Surgical approach, *n* (%)		
Open surgery	35 (26.7)	131
Conversion	9 (6.7)	
Laparoscopy	87 (64.4)	
Bleeding (mL)		
Mean [95% CI]	100 [100;200]	93
Blood units transfused, *n* (%)		
0	114 (92.7)	
1	5 (4.1)	125
2	4 (3.3)	
Resection of other organs, *n* (%)		
Bladder	2 (15.4)	13
Other	11 (84.6)	
Histological type, *n* (%)		
Conventional adenocarcinoma	122 (93.9)	130
Mucinous adenocarcinoma	6 (4.6)	
Undifferentiated carcinoma	1 (0.8)	
Adenosquamous carcinoma	1 (0.8)	
T, *n* (%) **		
1	9 (7.1)	126
2	22 (17.5)	
3	78 (63.9)	
4	17 (13.5)	
N, *n* (%) **		
1	58 (70.7)	82
2	24 ()29.3	
Absence of residual tumor, *n* (%)	39 (100.0)	39
Adjuvant CT, *n* (%)	99 (75.6)	131

** Colorectal cancer stage according to the 8th edition of the tumor, node and metastases (TNM) classification system [24]. CI: Confidence interval; CT: Chemotherapy.

**Table 4 cancers-16-01676-t004:** Postoperative surgical complications associated with liver resection.

Type of Complications	Overall Study Sample *N* = 150	Centers < 50 Hep/Year *N* = 63	Centers ≥ 50 Hep/Year *N* = 87	*p*	*N*
Overall morbidity *, *n* (%)	39 (26.0)	17 (27.0)	22 (25.3)	0.964	150
Clavien-Dindo CCI, *n* (%)					
I	5 (12.8)	2 (11.8)	3 (13.6)	0.907	39
II	17 (43.6)	9 (52.9)	8 (36.4)		
IIIa	11 (28.2)	5 (29.4)	6 (27.3)		
IIIb	4 (10.3)	1 (5.9)	3 (13.6)		
IV	1 (2.6)	0 (0.0)	1 (4.6)		
V	1 (2.6)	0 (0.0)	1 (4.6)		
Haemorrhage, *n* (%)	17 (11.3)	4 (6.3)	13 (14.9)	0.426	17
Grade A	15 (88.2)	3 (75.0)	12 (92.3)		
Grade B	2 (11.8)	1 (25.0)	1 (7.7)		
Liver failure, *n* (%)	21 (14.0)	9 (14.3)	12 (13.8)	0.530	21
Grade A	15 (71.4)	8 (88.9)	7 (58.3)		
Grade B	5 (23.8)	1 (11.1)	4 (33.3)		
Grade C	1 (4.8)	0 (0.0)	1 (8.3)		
Biliary fistula, *n* (%)	13 (89.0)	6 (9.8)	7 (8.2)	0.968	146
Bilioma, *n* (%)	8 (5.5)	3 (5.0)	5 (5.9)	1.000	145
Intra-abdominal abscess, *n* (%)	9 (6.2)	1 (1.7)	8 (9.4)	0.081	145
Reintervention, *n* (%)	5 (3.4)	1 (20.0)	4 (80.0)	1.000	148
Percutaneous drainage	1 (0.7)	0 (0.0)	1 (25.0)		
Surgical reintervention	4 (2.7)	1 (100.0)	3 (75.0)		
Medical complications, *n* (%)	15 (10.0)	8 (12.7)	7 (8.1)	1.000	15
Cardiac arrest	1 (6.7)	0 (0.0)	1 (14.3)		
Septic shock	2 (13.3)	1 (12.5)	1 (14.3)		
VTE or PE	1 (6.7)	1 (12.5)	0 (0.0)		
Other	11 (73.3)	6 (75.0)	5 (71.4)		
ICU stay (days)					
Mean [95% CI]	1.00 [0.00;2.00]	1.0 [1.0 to 2.0]	1.0 [1.0 to 2.0]	0.529	145
LOS (days)					
Mean [95% CI]	6.00 [4.00;8.25]	6.0 [4.0 to 8.0]	5.0 [4.0;9.0]	0.897	149
Re-admission, *n* (%)	16 (10.7)	5 (7.94%)	11 (12.6%)	0.513	150

* Number of patients who experience at least one postoperative complication. Hep: Hepatectomy; CCI: Comprehensive Complication Index; VTE: Venous thromboembolism; PE: Pulmonary embolism; ICU: Intensive care unit; CI: Confidence interval; LOS: Length of hospital stay.

**Table 5 cancers-16-01676-t005:** Postoperative surgical complications associated primary tumor surgical procedure.

Type of Complications	Overall Study Sample *N* = 134	*N*
Overall morbidity *, *n* (%)	35 (26.1)	35
Clavien-Dindo CCI, *n* (%)		
I	5 (14.3)	35
II	14 (40.0)	
IIIa	4 (11.4)	
IIIb	10 (28.6)	
IV	2 (5.7)	
V	0 (0.0)	
Reinterventions, *n* (%)		
Surgical reinterventions	9 (6.9)	130
Medical complications, *n* (%)	8 (6.2)	130
Description of medical complications, *n* (%)		
Septic shock	1 (12.5)	8
VTE or PE	2 (25.0)	
Other	5 (62.5)	
LOS, days		
Mean [95% CI]	7.00 [5.00;10.0]	130
Re-admission, *n* (%)	8 (6.2)	130
Reason for readmission, *n* (%)		
Anastomosis stricture	1 (12.5)	8
Paralytic ileus	1 (12.5)	
COVID-19	1 (12.5)	
Colorectal anastomosis dehiscence	1 (12.5)	
Pain	1 (12.5)	
Evisceration	1 (12.5)	
Rectal bleeding	1 (12.5)	
Portal thrombosis	1 (12.5)	

* Number of patients who experience at least one postoperative complication. CCI: Comprehensive Complication Index; VTE: Venous thromboembolism; PE: Pulmonary embolism; CI: Confidence interval; LOS: Length of hospital stay; COVID-19: Coronavirus disease 2019.

**Table 6 cancers-16-01676-t006:** A comparison of the liver-first approach (LFA) feasibility between the current study and the available evidence.

Study	Year *	Design	Number of Patients Starting Protocol	Number of Patients Completing Protocol	Feasibility (%)
Brouquet et al. [35]	2010	Retrospective	41	27	65.9%
Ayez et al. [36]	2013	Retrospective	42	31	74%
Sturesson et al. [37]	2017	Retrospective	75	49	65.3%
Wang et al. [32]	2016	Retrospective	18	16	88.9%
Mentha et al. [38]	2008	Prospective	35	30	85.7%
Verhoef et al. [39]	2009	Retrospective	23	17	73.9%
de Jong et al. [40]	2011	Prospective	22	16	72.7%
Kardassis et al. [41]	2014	Prospective	11	4	36.4%
Labori et al. [42]	2017	Retrospective	45	40	88.9%
Total	N.A.	N.A.	312	230	73.7%
Current study	---	Prospective	150	134	89.3%

* Year of publication. N.A.: Not available.

## Data Availability

Reported results can be fount at https://clinicaltrials.gov/study/NCT04683783?term=serradilla&rank=3 (accessed on 6 December 2023).

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
