# Peer review of "Feasibility and Short-Term Outcomes in Liver-First Approach: A Spanish Snapshot Study (the RENACI Project)"

_cancers, 2024, doi:10.3390/cancers16091676_

Round 1
Reviewer 1 Report (New Reviewer)
Comments and Suggestions for Authors
This paper, "Feasibility And Short-Term Outcomes In Liver-First Approach: A Spanish Snapshot Study (The RENACI Project)," presents valuable findings from a prospective observational study on the liver-first approach for patients with colorectal cancer and synchronous liver metastases in Spain. The study, conducted across 40 hospitals between June 1, 2019, to August 31, 2020, included 2288 hepatectomies, 150 of which utilized the liver-first approach. It aims to assess feasibility and short-term outcomes due to the dearth of comparative randomized clinical trials or prospective series. Notably, 89.3% of patients completed the treatment regimen.
Points to address include:
1. The comprehensive outcomes of the study, including resections of primary tumors alongside hepatectomies, is a significant contribution. Nevertheless, the multi-institutional nature brings limitations, such as the lack of specific numbers for unresectable or borderline resectable cases and a clear definition for borderline resectable. It is unclear whether extrahepatic metastases were included.
2. The paper mentions chemotherapy duration but not regimen specifics. Information on the administration rate of targeted therapy would also be beneficial.
3. With 89.3% of patients completing the treatment regimen, it's significant that 16 could not undergo primary tumor resection, 6 due to liver disease progression. The study does not indicate if any were eligible for repeat hepatectomy. Liver-first approach is apt for high tumor burden cases, often linked to early recurrence, and it's plausible for repeat hepatectomy in localized liver recurrence before primary tumor resection.
4. In the liver-first approach discussion, a noted issue is considering hepatectomy as the feasibility starting point rather than the onset of preoperative treatment, such as NAC. Hence, while the 89% completion rate aligns with previous reports, its significance is debatable. Simultaneous resection tends to have a high completion rate but carries increased complication risks. Safety-focused strategies are suggested for facilities with fewer than 50 annual liver resections.
5. The percentage descriptions in Table 4 are convoluted and require a clear explanatory text for better comprehension.
Comments on the Quality of English LanguageNo problems with the quality of English.
Author Response
Dear reviewer,
Thank you very much for the opportunity to make changes to the manuscript. We appreciate the time and effort you and each of the reviewers have dedicated in providing insightful feedback on ways to strengthen our paper. We have revised the manuscript in accordance with your suggestions. Our changes have been highlighted in the revised manuscript.
Our point-by-point responses to your comments and the revisions made are summarized below in the text. Your comments helped us considerably in improving our manuscript.
This paper, "Feasibility And Short-Term Outcomes In Liver-First Approach: A Spanish Snapshot Study (The RENACI Project)," presents valuable findings from a prospective observational study on the liver-first approach for patients with colorectal cancer and synchronous liver metastases in Spain. The study, conducted across 40 hospitals between June 1, 2019, to August 31, 2020, included 2288 hepatectomies, 150 of which utilized the liver-first approach. It aims to assess feasibility and short-term outcomes due to the dearth of comparative randomized clinical trials or prospective series. Notably, 89.3% of patients completed the treatment regimen.
Thank you very much indeed for this comment, we highly appreciate it.
The comprehensive outcomes of the study, including resections of primary tumors alongside hepatectomies, is a significant contribution. Nevertheless, the multi-institutional nature brings limitations, such as the lack of specific numbers for unresectable or borderline resectable cases and a clear definition for borderline resectable. It is unclear whether extrahepatic metastases were included.
Thank you very much for this comment. We agree with the reviewer that, unlike what may occur in pancreatic head cancer, there is no clear definition for borderline resectable liver metastases from colorectal cancer, which is often given by the technical capacity of the surgeon. In a multicenter study with 40 centers involved, it is impossible not to find differences in this sense, which can be a limitation.
Patients with extrahepatic disease were excluded. It has been added in line 195.
The paper mentions chemotherapy duration but not regimen specifics. Information on the administration rate of targeted therapy would also be beneficial.
Thank you very much for your comment. The neoadjuvant chemotherapy regimens have been included in Table 1. Similarly, the following sentence has been included on line 274:
All patients received neoadjuvant chemotherapy, the majority based on the FOLFOX and FOLFIRI regimens in combination with monoclonal antibodies (55.3%), with a mean of 6 cycles.
With 89.3% of patients completing the treatment regimen, it's significant that 16 could not undergo primary tumor resection, 6 due to liver disease progression. The study does not indicate if any were eligible for repeat hepatectomy. Liver-first approach is apt for high tumor burden cases, often linked to early recurrence, and it's plausible for repeat hepatectomy in localized liver recurrence before primary tumor resection.
Thank you very much for your comment. As the reviewer suggests, it is plausible repeating hepatectomy in localized liver recurrence before primary tumor resection. However, the optimal treatment strategy for patients with recurrent tumors has remained unclear.
In the current study, none of the surgeons considered the possibility of repeating the hepatectomy.
In the liver-first approach discussion, a noted issue is considering hepatectomy as the feasibility starting point rather than the onset of preoperative treatment, such as NAC. Hence, while the 89% completion rate aligns with previous reports, its significance is debatable. Simultaneous resection tends to have a high completion rate but carries increased complication risks. Safety-focused strategies are suggested for facilities with fewer than 50 annual liver resections.
Thank you very much for your comment. Table 4 compares the incidence of postoperative complications associated with liver resection between centers performing < 50 and ≥ 50 hepatectomies/year. There were no significant differences between them.
Nevertheless, we have included the following paragraph in line 388:
Although our study did not find significant differences between the centres performing < 50 and ≥ 50 hepatectomies/year. Simultaneous resection tends to have a high completion rate but has been associated with heightened risks of complications [4]. Therefore, safety-centered approaches would be recommended for facilities performing fewer than 50 liver resections annually.
The percentage descriptions in Table 4 are convoluted and require a clear explanatory text for better comprehension.
Thank you very much for your comment. The table has been modified.
Reviewer 2 Report (New Reviewer)
Comments and Suggestions for Authors
Comments and suggestions for Authors:
This study was aimed to show short-term results of liver first approach in patients of colorectal cancer with simultaneous liver metastases in multi-centers. There were some points of view to be clarified in this manuscript as shown below.
1, In 40 hospitals, hepatectomy for colorectal liver metastases were undergone in 1350 patients. Among them 150 (11%) patients underwent the Liver-First approach. It was not clearly shown how liver first approach was selected in each patient. Authors should clarify selection criteria of the liver first approach in each hospital. Was there a similar selection criteria?
2, Authors compared the short term results of liver first approach between hospital of >50 hepatectomies a year and <50 hepatectomies a year. Why did authors compare the short-term results between these two groups? Were there any difference in surgical morbidity rates and mortality rates after hepatectomies between these two groups? If there were some differences, it should be shown in this study subjects.
3, In this study liver first approach was completed perfectly in 89.3% of patients. What were reasons
for failure of liver first approach in this study? The reasons for failure of liver first approach should be clarified in 10.7% of patients who were tried to do liver first approach clearly by authors.
4, In this study, nineteen (12.8%) patients underwent tw-stage hepatectomy with portal vein embolization. Was there any difference in completion of liver first approach between the two-stage hepatectomy group and the one-stage hepatectomy group? It should be shown in the manuscript.
5, In this study group 89% of hepatectomies for colorectal liver metastases were undergone with primary-first approach or simultaneous resection approach in total hospital series. Why did authors not compare short-term results among these three different approaches?
Author Response
Dear reviewer,
Thank you very much for the opportunity to make changes to the manuscript. We appreciate the time and effort you and each of the reviewers have dedicated in providing insightful feedback on ways to strengthen our paper. We have revised the manuscript in accordance with your suggestions. Our changes have been highlighted in the revised manuscript.
Our point-by-point responses to your comments and the revisions made are summarized below in the text. Your comments helped us considerably in improving our manuscript.
In 40 hospitals, hepatectomy for colorectal liver metastases were undergone in 1350 patients. Among them 150 (11%) patients underwent the Liver-First approach. It was not clearly shown how liver first approach was selected in each patient. Authors should clarify selection criteria of the liver first approach in each hospital. Was there a similar selection criteria?
Thank you very much for your comment. All the centers followed the same inclusion criteria. Likewise, in a multicenter study of these characteristics, without a specific definition of what is unresectable or borderline resectable, there may be disparate criteria in this sense depending on the experience of the surgical team, which represents a limitation. In this sense there may be a slight discrepancy that is impossible to eliminate.
The following sentence was added in line 197:
Each participating center meticulously adhered to these inclusion criteria throughout the study.
In the same way, we added the following sentence in line 420:
Likewise, in a multicenter study of these characteristics, without a specific definition of what is unresectable or borderline resectable, there may be disparate criteria in this sense depending on the experience of the surgical team, which represents another limitation.
Authors compared the short-term results of liver first approach between hospital of >50 hepatectomies a year and <50 hepatectomies a year. Why did authors compare the short-term results between these two groups? Was there any difference in surgical morbidity rates and mortality rates after hepatectomies between these two groups? If there were some differences, it should be shown in this study subjects.
Thank you very much for your comment. In this study we tried to identify if there are differences in morbidity and mortality between low-medium and high-volume centres of liver surgery. Table 4 compares the incidence of postoperative complications associated with liver resection between centers performing < 50 and ≥ 50 hepatectomies/year. There were no significant differences between them.
Regarding the cut-off defining high vs low-medium volume is different between studies and countries. One of the most accepted is 50, so we chose that cut-off.
In this study liver first approach was completed perfectly in 89.3% of patients. What were reasons for failure of liver first approach in this study? The reasons for failure of liver first approach should be clarified in 10.7% of patients who were tried to do liver first approach clearly by authors.
Thank you for your comment, we really appreciate it. In this study, the hepatic approach was completed in 100% of patients. It was not possible to perform colorectal cancer surgery in 16 (10.7%) patients (Table 3) for the following reasons: nine due to complications after liver surgery, 6 due to progression of liver disease, and 1 due to postoperative death (line 298). One-hundred and thirty-four (89.3%) patients completed the therapeutic regimen (neoadjuvant chemotherapy + liver resection ± chemotherapy/radiotherapy of the primary tumor + surgery of the primary tumor) (line 303).
In this study, nineteen (12.8%) patients underwent two-stage hepatectomy with portal vein embolization. Was there any difference in completion of liver first approach between the two-stage hepatectomy group and the one-stage hepatectomy group? It should be shown in the manuscript.
Thank you for your comment, we really appreciate it. Nineteen (12.8%) patients had undergone a previous portal vein embolization and underwent a 2-stage hepatectomy (line 278).
There were no differences in morbidity and mortality between the two-stage hepatectomy group and the one-stage hepatectomy group.
The following sentence was added in line 324:
There were no significant differences in the complication rates between patients with 2-stage hepatectomy and the rest of the patients (28.1 vs 24.3, p=0.682).
In this study group 89% of hepatectomies for colorectal liver metastases were undergone with primary-first approach or simultaneous resection approach in total hospital series. Why did authors not compare short-term results among these three different approaches?
Thank you very much for your comment. Throughout the study inclusion period, a total of 2,288 hepatectomies were performed in the study centres, 1,350 for CRLM, with a mean of 57.2 hepatectomies per centre (23 to 112). Among them, 150 (11.1%) patients had undergone a LFA and were included in the study (line 253).
On the other hand, it is true that most of the patients were operated on using a classic or simultaneous approach. However, we only have information on patients who underwent liver-first approach. This is a snapshot study, a cross-sectional study design which enables an actual insight into current practice by collecting data in a short period of time in a large number of centers and therefore creates greater generalizability than randomized controlled trials or longitudinal studies. Snapshot studies are based on collaborative research and supported by the Spanish Association of Surgeons. The aim of this Spanish snapshot study was to assess short-term postoperative outcomes after elective LFA an to offer a “picture” of the use of this approach in a real scenario.
This manuscript is a resubmission of an earlier submission. The following is a list of the peer review reports and author responses from that submission.
Round 1
Reviewer 1 Report
Comments and Suggestions for Authors
Thank you for the opportunity of reviewing this paper.
The authors should be congratulated for the leading a large multiinstitutional work.
It would be interesting to have some description about how the data was managed, want kind of information was recolected. It was probably described in detail elsewhere, but a short description would be useful. In particular, did you also put in the data from primary first approach (classic approach), which seems to be the majority of cases in Spain. If this data is available, I believe a comparative study would be more interesting. As you say, this is a large series about feasability, but in general, the surgical community already knows this is a feasable approach; what is not so clear is whether its better than the classic approach.
Also, I think it is important to include the data about de burden of disease in the liver: number of lesions, size, number of segments involved, etc. That gives an idea about your selection for this approach, and is data that is usually described in papers about this topic, and the type of liver resections I believe is not enough. Also, it would be interesting to see if centers with more hepatectomies get patients with more disease burden.
Author Response
Thank you for the opportunity of reviewing this paper. The authors should be congratulated for the leading a large multi institutional work.
Thank you very much for the comment, we highly appreciate it.
- It would be interesting to have some description about how the data was managed, want kind of information was recollected. It was probably described in detail elsewhere, but a short description would be useful. In particular, did you also put in the data from primary first approach (classic approach), which seems to be the majority of cases in Spain. If this data is available, I believe a comparative study would be more interesting.
Dear reviewer, we really appreciate your comment.
Regarding the variables, the following paragraph has been added in the methods section, line 213:
2.7. Study variables
The following variables were studied: age, sex, Body Mass Index, ASA grade, and past medical history; clinical symptoms; carcinoembryonic antigen (CEA) and carbohydrate antigen (CA) 19.9 preoperative levels; location of the primary tumour, number, size, and location of liver metastases, and extrahepatic disease; need for stent placement or colostomy, neoadjuvant chemotherapy, and time from diagnosis to start of chemotherapy; portal embolization, two-stage hepatectomy, type of surgery, major (greater than or equal to three segments) and minor (less than three segments) hepatectomy, operating time, approach, intraoperative blood loss, clamping time, R status, degree of tumour regression, postoperative morbidity and mortality (according to the Clavien–Dindo classification) [23], bile leak, post-hepatectomy insufficiency and haemorrhage defined by International Study Group of Liver Surgery classification [24–26], length of hospital stay (LOS), readmissions, adjuvant chemotherapy, and radiotherapy; number of patients with resection of the primary tumour, type of surgery, approach, operating time, intraoperative blood loss, postoperative morbidity and mortality after primary resection, LOS, readmissions, histological type, TNM classification, degree of tumour regression, and adjuvant chemotherapy; and postoperative follow-up (months), death, and recurrence.
On the other hand, it is true that most of the patients were operated on using a classic approach. However, we only have information on patients who underwent liver-first approach (LFA). This is a snapshot study, a cross-sectional study design which enables an actual insight into current practice by collecting data in a short period of time in a large number of centers and therefore creates greater generalizability than randomized controlled trials or longitudinal studies. Snapshot studies are based on collaborative research and supported by the Spanish Association of Surgeons. The aim of this Spanish snapshot study was to assess short-term postoperative outcomes after elective LFA an to offer a “picture” of the use of this approach in a real scenario.
- As you say, this is a large series about feasibility, but in general, the surgical community already knows this is a feasible approach; what is not so clear is whether it is better than the classic approach.
To the authors knowledge, there is no randomized clinical trials or prospective series comparing classical with liver-first approach (LFA). As the reviewer mentioned, current evidence suggests that there are not significant differences among sequential primary-first, LFA, or synchronous resection.
The current study was focus on describing the characteristics of the study sample and provided only preliminary results. Nevertheless, further analysis evaluating association of potential relevant clinicopathological factors with prognosis and determining the best candidates for LFA will be perform.
- Also, I think it is important to include the data about de burden of disease in the liver: number of lesions, size, number of segments involved, etc. That gives an idea about your selection for this approach, and is data that is usually described in papers about this topic, and the type of liver resections I believe is not enough. Also, it would be interesting to see if centers with more hepatectomies get patients with more disease burden.
Dear reviewer, we really appreciate your comment. These data have been added to table 1. We have also included the following sentence in line 268:
There were no differences between both groups in terms of preoperative location of liver metastases or bilobar involvement (30 patients [47.6%] in centres < 50 hepatectomies/year vs 55 patients [63.2%] in centres ≥ 50 hepatectomies/year, p=0.083).
Reviewer 2 Report
Comments and Suggestions for Authors
- Define second and third degree hospital for HBP surgery included in the study
- Line 188 - In case of locally advanced rectal tumors, radiotherapy or chemotherapy/radiotherapy is carried out, and finally surgery of the primary tumor is performed. – The authors should also discuss the possibility to perform LFA in the window of opportunity between the end of radiotherapy and the actual operation for resection of the rectal tumor that is due 10-12 weeks later.
Line 346 - The fact that in this shows approximately 50% of patients did not have a rectal tumor suggests that this liver-first strategy is expanding its indications – however I think is valuable to discuss in more detail the role of LFA especially in patients with rectal cancer
Comments on the Quality of English LanguageMinor revision required
Author Response
- Define second and third degree hospital for HBP surgery included in the study.
The following information was added in the methods section, line 170:
A total of 40 second (area hospitals with approximately 500 beds, and on average of 270 specialists and 50 residents) and third-level (university reference hospitals with approximately 800-1000 beds, on average of 680 specialists and 300 residents, and great teaching intensity) hospitals decided to participate in the study.
- In case of locally advanced rectal tumors, radiotherapy or chemotherapy/radiotherapy is carried out, and finally surgery of the primary tumor is performed. – The authors should also discuss the possibility to perform LFA in the window of opportunity between the end of radiotherapy and the actual operation for resection of the rectal tumor that is due 10-12 weeks later.
We really appreciate your comment. Thank The following paragraph was added in the discussion section, line 402:
Finally, it should be mentioned that despite LFA strategy prioritises the removal of metastases, it still includes a chemotherapy-free period of at least 3 months after liver surgery [6,7]. It has been recently proposed a new LFA strategy that proposed resection of the liver metastases during the interval between long-course chemoradiation and rectal cancer surgery [44]. The authors reported that 87.5% of patients successfully underwent the liver first strategy and underwent both liver and rectal treatment [45]. These results are similar to those found in our study, with the particularity that our study included 150 cases and the study by Bonnet et al [44] only 24 patients.
Nevertheless, this strategy offers interesting possibilities that must be analysed in future studies with a larger number of cases.
- The fact that in this shows approximately 50% of patients did not have a rectal tumor suggests that this liver-first strategy is expanding its indications – however I think is valuable to discuss in more detail the role of LFA especially in patients with rectal cancer.
Dear reviewer, thank you very much for your consideration. The following paragraph was added in the discussion section, line 394:
However, LFA has preferentially been applied to patients with rectal tumours and high liver tumour burden [6-9]. In patients with CRC and liver metastases, both resections can be performed in a single procedure [9]. Interestingly, this strategy did not have better survival outcomes, while was associated with more complications [4]. Current evidence suggests that in patients with CRC, LFA is not inferior to other approaches in patients with unilobar SCRLM [8,9]. Nevertheless, LFA was associated with a clear survival ad-vantage over both the primary-first and simultaneous approaches in patients with multiple bilobar metastases [8,9].
Reviewer 3 Report
Comments and Suggestions for Authors
I have read with interest this study that concerns the feasibility of the liver first approach in a nationwide Spanish cohort. While the topic is of interest, and while the paper is overall well written, it contains some major flaws that strongly invalidate the conclusions
1. Please clarify your study endpoints. There is no mention in the paragraph 2.6 about the volume effects. That should be added and motivated. Yet, the cut-off of 50 cases per year should be justified.
2. More importantly, you are pretty much comparing apples with oranges. General and tumoral features of patients treated in low versus high volume centers are completely missing. Size, number and (importantly) locations/intrahep vascular contacts of liver mets should be reported to make the reader understand if those two centers operated on similar patients. Showing the CEA level and the use of neoadjuvant chemo is not enough.
3. The flow of the patients’ numbers is unclear. Please add a figure to show the patients’ flow. Looking at paragraph 3.3 and looking at table 2 and 3, the numbers are incorrect. For instance, at row 243, 87+54= 141. 9 patients are missing. Other numbers have the same problem in the text and in the tables.
4. In table 3 exploratory laparotomy is reported twice.
5. Complications may be related to the surgical attitude and technique. In a multicenter study, surgical attitude and technique are different. This represents a strong selection and interpretation bias. By the way, surgical technique, and postoperative care are not detailed at all.
6. Postop results should be compared with the other two strategies (bowel-first and simultaneous approach). Without such a comparison, your data are not interpretable.
7. You have 2.6% of mortality (in the overall group), which by the way should be defined if at 30- or at 90-day. This info is missing. Anyway, with such a good result, is difficult to find differences among groups unless you have a very large sample size. Unfortunately, 150 patients are not enough for that analysis. In this sense, a kind of "a priori" sample size calculation should be planned to make your expected results credible. In other words, your numbers have not enough power.
8. Six tables are a number. These tables show too many data, while at the same time some of the most important data are missing.
9. 60% of patients had minor resections. Were these resections minor but complex – meaning bilateral parenchymal sparing hepatectomies - or they were simple limited resections? Again, the liver tumor burden should be detailed otherwise all the comparisons are meaningless.
Author Response
I have read with interest this study that concerns the feasibility of the liver first approach in a nationwide Spanish cohort. While the topic is of interest, and while the paper is overall well written, it contains some major flaws that strongly invalidate the conclusions.
We hope that after the changes recommended by the reviewer, the article will be able to eliminate or at least reduce some of the important defects mentioned by the reviewer.
- Please clarify your study endpoints. There is no mention in the paragraph 2.6 about the volume effects. That should be added and motivated. Yet, the cut-off of 50 cases per year should be justified.
Thank you very much for you comment. The following sentences have been added in line 158:
On the other hand, there is an inverse relationship between hospital and surgeon volume and mortality in many types of complex surgery.
And line 208:
The secondary end-points were 90-day postoperative morbidity including liver and colorectal surgery (all type of postoperative complications), and to investigate the volume effect on outcomes this complex surgery.
Regarding he cut-off defining high vs low volume is different between studies and countries. One of the most accepted is 50, so we chose that cut-off.
- More importantly, you are pretty much comparing apples with oranges. General and tumoral features of patients treated in low versus high volume centers are completely missing. Size, number and (importantly) locations/intrahep vascular contacts of liver mets should be reported to make the reader understand if those two centers operated on similar patients. Showing the CEA level and the use of neoadjuvant chemo is not enough.
Dear reviewer, we really appreciate your comment. These data have been added to table 1. We have also included the following sentence in line 268:
There were no differences between both groups in terms of preoperative location of liver metastases or bilobar involvement (30 patients [47.6%] in centres < 50 hepatectomies/year vs 55 patients [63.2%] in centres ≥ 50 hepatectomies/year, p=0.083).
- The flow of the patients’ numbers is unclear. Please add a figure to show the patients’ flow. Looking at paragraph 3.3 and looking at table 2 and 3, the numbers are incorrect. For instance, at row 243, 87+54= 141. 9 patients are missing. Other numbers have the same problem in the text and in the tables.
We really appreciate the thorough review of the data. Regarding the comment: "For instance, at row 243, 87+54= 141. 9 patients are missing", if you would be so kind as to review table 2, the "Surgical approach" section you will see that there is another concept with 9 cases "Conversion". Furthermore, in all cases the number of subjects analyzed has been stated.
- In table 3 exploratory laparotomy is reported twice.
It has been corrected.
- Complications may be related to the surgical attitude and technique. In a multicenter study, surgical attitude and technique are different. This represents a strong selection and interpretation bias. By the way, surgical technique, and postoperative care are not detailed at all.
The following limitation was included, line 413:
As this is a multicenter study, there may be some differences between the surgical techniques between the different centers and may influence surgical outcomes. However, we defined clearly the standard procedure and the limits on acceptable technical variation.
- Postop results should be compared with the other two strategies (bowel-first and simultaneous approach). Without such a comparison, your data are not interpretable.
The data mentioned by the reviewer are shown in tables 4 and 5. They have been shown separately to facilitate their reading.
- You have 2.6% of mortality (in the overall group), which by the way should be defined if at 30- or at 90-day. This info is missing. Anyway, with such a good result, is difficult to find differences among groups unless you have a very large sample size. Unfortunately, 150 patients are not enough for that analysis. In this sense, a kind of "a priori" sample size calculation should be planned to make your expected results credible. In other words, your numbers have not enough power.
Statistical significance does not refer to the mortality rate, but rather to the total data from the Clavien-Dindo classification.
As mentioned in the methodology section, one of the secondary end-point was 90-day postoperative morbidity including liver and colorectal surgery (all type of postoperative complications).
The overall morbidity rate was 26% (39/150).
- Six tables are a number. These tables show too many data, while at the same time some of the most important data are missing.
Thank you very much for your consideration. The data suggested by the reviewer have been included in the corresponding tables.
- 60% of patients had minor resections. Were these resections minor but complex – meaning bilateral parenchymal sparing hepatectomies - or they were simple limited resections? Again, the liver tumor burden should be detailed otherwise all the comparisons are meaningless.
Dear reviewer, thank you very much for your comment. In many cases it is difficult to establish the tumour burden. We have tried to summarize all this information in Table 1 (distribution of liver metastases) and Table 2 (type of surgery performed). Unfortunately, we do not have information about the vascular relationship of the lesions.
Round 2
Reviewer 3 Report
Comments and Suggestions for Authors
I thank the authors for the effort in providing this R1 version of their manuscirpt together with the reviewer's reply. Now the paper has been improved, but some major flaws remain. In details:
1. I am still confused about your comparison. Thank for adding the location of CLM in table 1 and the types of surgery in table 2. However, the sum of the types of surgery reported in table 2 is superior to the total of patients included. Yet, there is not much correspondence between the location of tumors reported in table 1 and the supposted correspondent operations detailed in table 2.
2. What's the meaning of NPLM reported in table 1?
3. My previous comment raised at #7 was referred to the fact the you should have a sufficient large sample size to make statistical inference on a given variable. I still believe that with a total of 150 patients most of your considerations are not really data-driven.
4. To support the feasibility of the LFA you should show the data and results of the bowel-first and of the simultaneous approaches. Since these other two patients' cohorts are here missing you are just offering a picture of the results of one treatment over three total possibilities. This is a limitation, which is acceptable, that must be clearly stated. Yet, no definitive conclusions on feasibility of LFA can be done with your data.
5. Please remove/change the statements "this is the first multicentre clinical study to evaluate the feasibility of the LFA [...]". This is not the case. Since you quoted some of the previous, much more larger, studies on the same topic (i.e. reference n. 8 and others), your study cannot be the first in line.